# Using Microorganismal Consortium and Bioactive Substances to Treat Seeds of Two Scots Pine Ecotypes as a Technique to Increase Re-Afforestation Efficiency on Chalk Outcrops

Vladimir M. Kosolapov [1], Vladmir I. Cherniavskih [1,2], Elena V. Dumacheva [1,2], Luiza D. Sajfutdinova [1,2], Alexander A. Zhuchenko, Jr. [2], Alexey P. Glinushkin [2], Helena V. Grishina [2], Valery P. Kalinitchenko [2,*], Svetlana V. Akimova [2,3], Natalia A. Semenova [4], Leonid V. Perelomov [5] and Svetlana V. Kozmenko [5,6]

1 Federal Williams Research Center of Forage Production and Agroecology, Nauczny Gorodok 1, 141055 Lobnya, Russia; kormoproizvodstvo@yandex.ru (V.M.K.); cherniavskih@mail.ru (V.I.C.)
2 All-Russian Research Institute of Phytopathology, 143050 Moscow, Russia; ecovilar@mail.ru (A.A.Z.J.); glinale1@mail.ru (A.P.G.); opmogreen@mail.ru (H.V.G.)
3 Institute of Horticulture and Landscape Architecture, Russian State Agrarian University—Moscow Timiryazev Agricultural Academy, 127434 Moscow, Russia
4 Federal Scientific Agroengineering Center VIM, 1st Institutskiy Proezd, 109428 Moscow, Russia; natalia.86@inbox.ru
5 Laboratory of Soil Chemistry and Ecology, Faculty of Natural Sciences, Tula State Lev Tolstoy Pedagogical University, Lenin Ave., 125, 300026 Tula, Russia; kozmenko@sfedu.ru (S.V.K.)
6 Academy of Biology and Biotechnology, Southern Federal University, 105/42 Bolshaya Sadovaya str., 344006 Rostov-on-Don, Russia
* Correspondence: kalinitch@mail.ru

**Abstract:** The present research is focused on various pine ecotypes' seed reproduction in the chalky substrate, challenging environmental conditions on the carbonate soils on chalk outcrops in the south of the Central Russian Upland in relation to pine woods re-afforestation. The winter and spring sowing methods were studied, along with a pre-seeding treatment, by biopreparations based on a consortium of *Glomales* fungi, bacteria of the genus *Bacillus*, and bioactive substances. The seeds of two pine ecotypes, *Pinus sylvestris* L.; *Pinus sylvestris* var. *cretacea* Kalenicz exKom, underwent treatment. The study revealed that biopreparations and bioactive substances promote higher pine seed germination rates and ensure the stability and survivability of seedlings in an environment that is unfavorable for plant and tree ontogenesis. Applying biopreparations proved effective during spring sowing, whereas, in the case of winter sowing, their positive impact was not statistically significant. The net effect size of the three organized factors studied in the experiment (pine ecotype, biopreparation, sowing term) ($h^2_x$) on the "survivability of *P. sylvestris* seedlings" effective feature significantly increased from 90.8 to 93.8%. The effect size of the "pine ecotype" factor on seedling survivability in *P. sylvestris* was at its highest (14.4%) during the seedlings' first-year growth period. The effect size of the "sowing term" factor was at its highest (79.4%) at the stage of seed germination. The effect size of the "biopreparation" factor was at its highest (44.0%) during the seedlings' second-year growth stage. Our results indicate that it is preferable to create forest plantations on chalk outcrops using *Pinus sylvestris* var. *cretacea* ecotypes and with pre-sowing seed treatment via biopreparations based on a microorganismal consortium and Biogor KM. The Spearman correlation between the nitrification capacity of soil substrate and seedling survivability during the first three growth periods (from planting till the next year's springtime) was of a moderate size ($r_s = 0.617$–$0.673$, $p < 0.05$). To improve the growth and productivity of young and mature Scots pine stands, a Biogeosystem Technique (BGT*) methodology was developed.

**Keywords:** *Pinus sylvestris* var. *cretacea* Kalenicz. ex Kom.; *Pinus sylvestris* L.; artificial forest restoration (re-afforestation) by seed; sowing terms; biopreparations; microorganismal consortium; bioactive substances; degraded soils; biogeosystem technique

## 1. Introduction

At present, the environmental and biological function of forests in land sustainability and water conservation is vital for the well-being of human society. Currently, the vulnerability of growing forest ecosystems badly influences both the climate and anthropogenically disturbed lands [1–3].

The afforestation of slopes and abandoned and degraded farmlands provide sustainable ecosystem development. This is an important tool in land resource management, increasing biodiversity, $CO_2$ sequestration, soil carbon content (as a part of humus), and preventing the uncontrolled transfer of contaminants within an ecosystem [4–8]. Arboriculture on low-productivity lands that are unsuitable for farming, as well as forest restoration and the re-introduction of forest species, is of a top priority in the economical and recreational sphere of societal development [9–12].

In Russia, re-afforestation and forest improvement are underway. One of the large-scale regional measures to restore cleared and recover thinned stands and improve water conservation zones and water objects was the "Green Capital" project launched in the Belgorod region in 2010. Its goal was to create a comfortable environment for residents living in the Belgorod region [13].

Since the implementation of this project, over one-hundred-thousand hectares of forest plantations have been planted on chalky slopes and erosion-hazardous sites, and by 2020, forestation in the region had reached 15%. Since the project has been proven to be effective, the relevant activities continued. For afforestation, both ash-leaved species (*Robinia pseudoacacia* L.; *Betula pendula* Roth.; *Fraxinus excelsior* L.; *Acer negundo* L.; *Ulmus laevis* Pall.; *Populus alba* L.) and indigenous woody and shrubby species (*Quercus robur* L.; *Caragana arborescens* Lam.; *Lonicera xylosteum* L.; *Sambucus racemosa* L.; *Pinus sylvestris*, etc.) have been used. Special attention has been paid to the restoration of *Pinus sylvestris* plantation areas [14].

In the course of implementation, a problem has emerged—afforestation of low-productivity and degraded lands located on sites in complicated landscapes and ravine complexes with chalk outcrops [15]. This challenge requires particular scientific and technological solutions. An example of low-productivity and degraded lands with chalk outcrops in the south of the Central Russian Upland is shown in Figure 1.

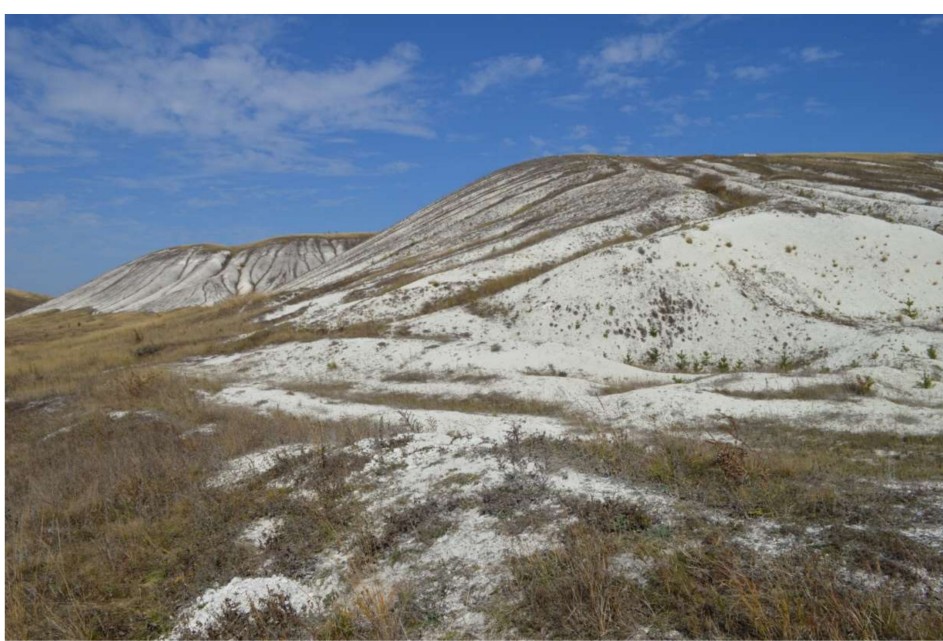

**Figure 1.** Low-productivity degraded lands with chalk outcrops in the south of the Central Russian Upland (photo by V.I. Chernyavsky).

The unfavorable forest growth conditions of a chalky outcrop substrate are an obstacle to the formation of a continuous vegetation cover. The hard and rocky chalky substrate, barely touched by soil formation processes, is found to possess some adverse properties, i.e., it has a strict light and water regime, high albedo, and high physiological dryness [16–18]. This kind of chalky outcrop substrate is suitable for the growth of an insignificant number of highly stenoecious organisms—endemic and specific in terms of nutritional adaptation species (Figure 2) [19–21].

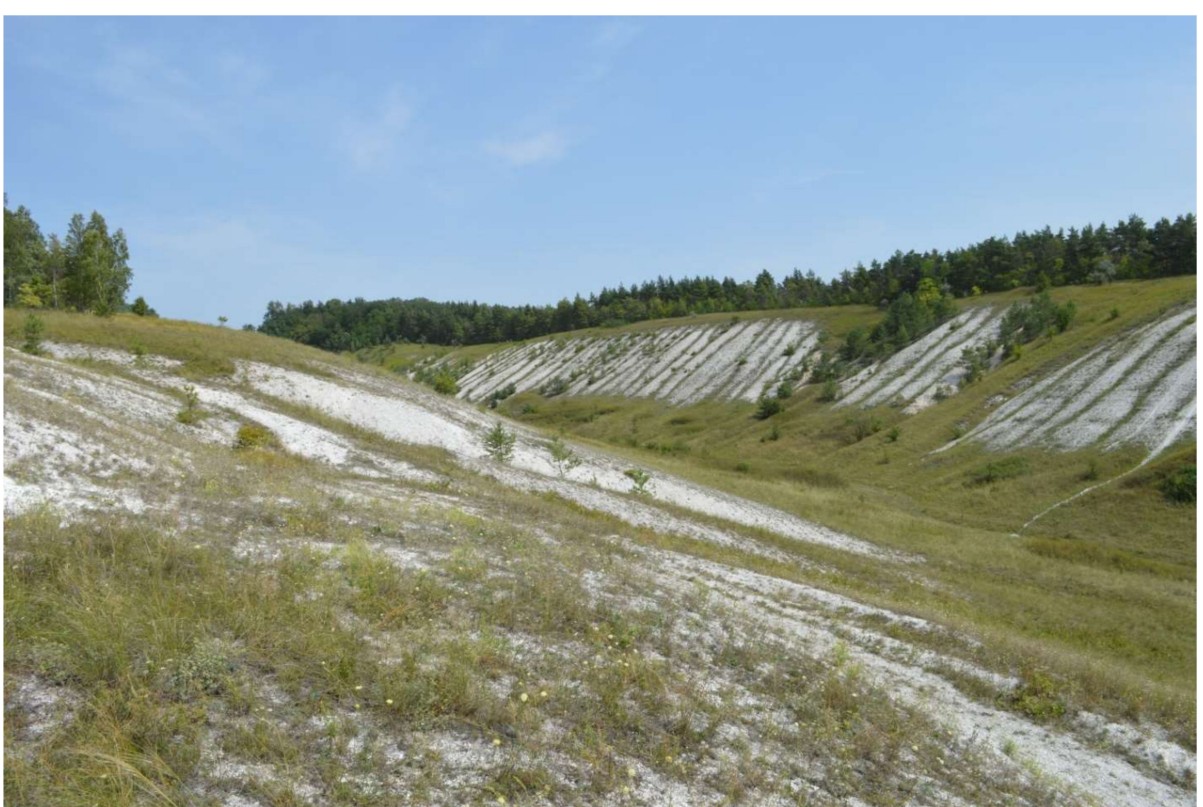

**Figure 2.** Seed restoration of Scotch pine on a chalk outcrop in the south of the Central Russian Upland, Belgorod Region (Photo by V.I. Chernyavsky).

Landscape complexes with carbonate rock outcrops are quite widespread worldwide; calcareous outcrops in the steppe, semi-desert, and desert zones of Eurasia hold a special place [22,23].

*Pinus sylvestris* L. is used for the creation of forest plantations in the complex landscape conditions of the carbonate soils on the chalk outcrops. Two main *Pinus sylvestris* ecotypes grow in the south of the Central Russian Upland. The *P. sylvestris* L. ecotype is the most widely used in cultivation. A second ecotype, *Pinus sylvestris var. cretacea* Kalenicz., is an indigenous relict species that grows on chalk outcrops. The authors of this study examined both ecotypes in order to identify the species that is the most promising for cultivation on Cretaceous outcrops. A detailed description of the ecotypes is given below.

For calcareous landscapes of southern Central Russian Upland, *Pinus sylvestris*, also known as common Scots or Scotch pine (Figure 2), has a special meaning. Thanks to the plant's unique morphological and biological traits and its ability to grow on poor soils, pine plantations are indispensable with regard to forest re-cultivation in unfavorable soil conditions [24–26].

This tree, being one of the most widespread conifers in the Northern Hemisphere, with areas ranging from Western Europe to Eastern Asia, has over 150 morphological varieties and eco-groups. This species has the widest environmental range and is the most promising crop for establishing forests on carbonate substrates. The species *P. sylvestris*

is polymorphic and, depending on the soil and climatic conditions, can form different ecotypes [27,28]. The species is known to have a complicated glacial and post-glacial history in the region, affected by both recent global climate change and human activities. Comprehensive surveys of this aspect are available [29].

The reasons for the development of re-afforestation techniques on chalk outcrops based on the *P. sylvestris* species are two-fold:

Firstly, to establish sustainable forest plantations in complicated soil and landscape conditions and to master technological approaches in using *P. sylvestris* for re-afforestation in various regions;

Secondly, for the re-introduction and restoration of relic forests, which once existed in the south of the Central Russian Upland.

In the Last Glacial Maximum, the pine survived in small spotted ice-free refugia [30,31].

It is of great importance that, in the Last Glacial Maximum, in the extraglacial zone of southern Central Russian Upland, pine forests (calcareous woods) based on the special *Pinus sylvestris* var. *cretacea* pine ecotype were quite widespread [32,33]. Presently, they are preserved in several refugia in Russia and Ukraine [34–36]. Usually, these places are very hard to reach (Figure 3).

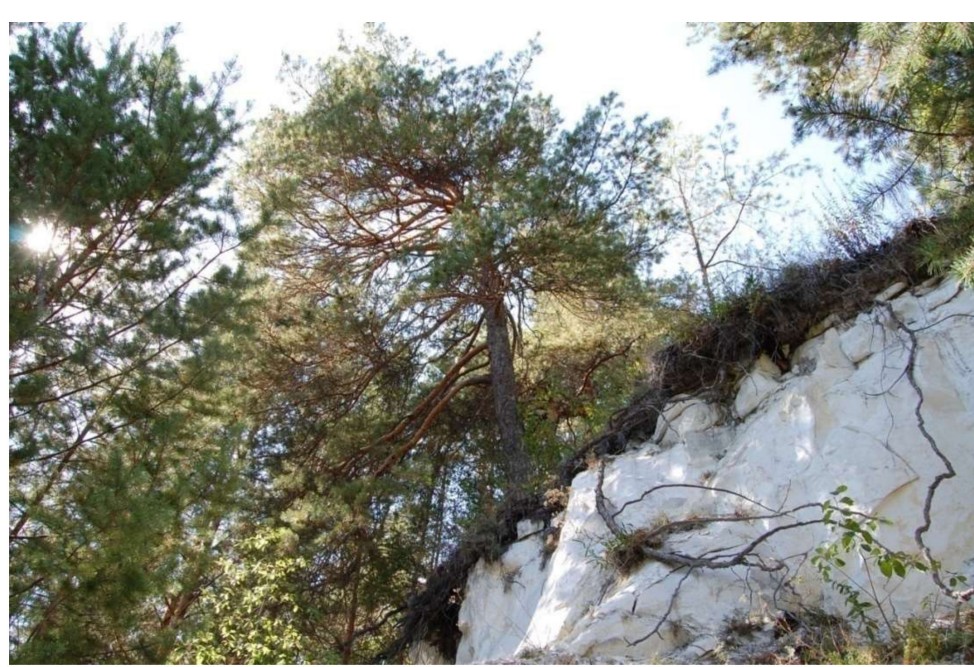

**Figure 3.** *Pinus. Sylvestris* var. *Cretacea* Kalenicz. exKom habitat in the Belgorod region (photo by V.I. Chernyavsky).

The use of this pine ecotype could facilitate the sustainable establishment of forest plantations on calcareous outcrops. Researchers have considered the potential of selecting resistant ecotypes in chalk pine habitats for future investigations of their development and dispersal throughout the area. A special direction of research is the restoration of relict chalk woods by sowing seeds and planting seedlings of *P. sylvestris* var. *cretacea* [15].

An important task is mastering and improving seed reproduction techniques and establishing conifer plantations by sowing seeds directly in hardly accessible areas, which can be cost-effective due to reductions in nursery handling and planting costs [37,38].

The ability of *Pinus* L. species to form symbiotic connections with beneficial soil microflora and arbuscular mycorrhiza (AM) is well-known [39]. This type of plant–microbe interaction is considered the most common and effective. Arbuscular mycorrhiza generates a diverse microbiocoenosis, which has a positive influence on the soil as an agent restoring soil fertility, enhancing the host plant's ability to absorb hardly available soil minerals,

increasing water intake, which promotes the cell growth by elongation and speeds up the growth of the host plant at the organismal level [40–42].

A cutting-edge research trend is to study the impact of biopreparations on *Pinus* species' capacity for survival and establishment [43–45].

Zygomycetes of the order *Glomales* have been shown to be promising for practical use [46]. Biopreparations containing these fungi were based on best practices in using bacterial associations with mycorrhizal fungi, bioactive substances, and macro- and micro-elements since, when used together, they display synergy and achieve a better performance of processes compared to pure cultures.

Positive outcomes following the application of biopreparations based on the consortium of *Glomales* fungi and bacteria of the genus *Bacillus* on various herbaceous plants and shrubs have been increasingly reported [47,48]. The effectiveness of biopreparations for better *Pinus* seed germination [49–51], as well as the capacity for survival and seedling development, has been demonstrated [52–54].

The present paper is concerned with the appraisal of forest restoration and regeneration approaches on the chalk outcrops in southern Central Russian Upland for two Scots pine ecotypes by sowing seeds and increasing its efficiency by applying biopreparations based on the consortium of *Glomales* fungi and *Bacillus* bacteria. To improve the young and mature Scotch pine stand growth and productivity, a Biogeosystem Technique (BGT*) methodology was developed.

## 2. Materials and Methods

### 2.1. Region of Investigations

The research was carried out in the slope area in the Central Russian Upland southwestern macroslope within the Belgorod region (Russia). The region has a high level of agricultural and industrial development. Thus, the degree of soil erosion is high in the region. The area is heavily indented by ravines and gullies ("balkas"). Against this backdrop, the most common landscapes in the region are those with eroded calcareous soils and chalk outcrops. Due to the high aridity, the soil environment is poorly adapted to the intensive agricultural production. These landscapes can be used for the establishment of forest plantations and Scots pine chalk woods renewal.

The climatic conditions in the region are diverse. The average annual temperature ranges from 5.4 °C to 6.7 °C. The average summer temperatures are from 8.4 °C to 19.6 °C. The average winter temperatures range from −6.5 °C to −8.0 °C. Annual rainfall differs from year to year and, on average, ranges from 530–550 mm in the northwest of the region to 465–490 in the southeast. In certain years, the amount of precipitation can decrease to 300 mm or increase up to 800 mm. During the vegetation period, 65%–75% of the total precipitation falls.

To begin the trial, seeds were selected in two locations that most resembled the conditions for Scots pine growth in European Russia's south: in the Titovskyi Bor terrain and Bekaryukovskyi Bor terrain.

The Titovskyi Bor terrain (50.383653° N; 36.824951° E) is located on the left bank of the Nezhegol river floodplain in the Shebekino district of the Belgorod region. Here, the pines grow on sandy soils of various origin. Pine plantations that are 70 years old are of artificial origin, with elements of natural self-renewal via spontaneous natural self-seeding. Taxonomically, these trees belong to the common Scots pine (*P. sylvestris*).

The Bekaryukovskyi Bor terrain (50.439334° N; 37.067669° E) is situated on the right high bank of the Nezhegol river, in the Shebekino district of Belgorod region. The pine plantation is of a natural origin, having been established in ravines on chalk outcrops and washed-off soils, which formed on chalk-based residual soil. The plantation appears to be formed of several surviving old trees, the remains of once-existing relict chalk wood with a tertiary relict species—the chalk (Cretaceous) pine. Taxonomically, it can be assigned to the ecotype of the common Scots pine *P. sylvestris* var. *cretacea*.

To collect cones, typical trees at the reproductive stage were selected. Seeds were collected in March 2018. Upon collection, the cones were dried and then threshed to remove seeds, which were then processed to reach sowing conditions (germination rate—90%, varietal purity—98%, humidity—7%). Before sowing, the seeds were stored in small canvas bags at room temperature.

### 2.2. Experiment Location and Conditions

The research was carried out on a chalk outcrop on the left bank of the Mandzhokhoga river (50.452995° N; 37.736746° E), a tributary of the Oskol river in the Volokonovka district of the Belgorod region. The slope was of a southeastern exposure. The soil was residual carbonate (calcimorphic) chernozem, greatly eroded on the chalk eluvium. The substrate's agrochemical properties in the 0–20 cm layer were as follows: humus content = 2.13%, pH = 7.84, $N_{total}$ = 0.16%., N-NO$_3$ content = 14.5 mg kg$^{-1}$, nitrifying capacity 17.9 mg kg$^{-1}$.

Weather conditions directly recorded during the field research are provided in Table 1.

**Table 1.** Weather conditions during the research period.

| Year | Rainfall, mm | | Average Annual Temperature, °C | | HTI | |
|---|---|---|---|---|---|---|
| | **In Fact** | **Normal** | **In Fact** | **Normal** | **In Fact** | **Normal** |
| 2018 | 693.6 | | 9.3 | | 1.15 | |
| 2019 | 429.2 | 553 | 10.4 | 6.3 | 0.59 | 1.0 |
| 2020 | 437.1 | | 9.2 | | 1.22 | |

HTI—Selyaninov's hydrothermal index, characterizing the water availability level in an area. It can be calculated using the following formula: $K = R \times 10/\Sigma t$, where R is the cumulative precipitation (rainfall) in mm for the period with air temperature above +10 °C, $\Sigma t$—cumulative temperature above +10 °C for the same period.

The total accounting area of the experimental plots was 42 m$^2$. The total area of the experiment was 168 m$^2$. In the model (a three-factor field experiment),the field germination rate and survivability of seedlings were tested over the course of two years for the seedlings of *P. sylvestris* and *P. sylvestris* var *cretacea*, depending on the sowing terms and seed treatment by biopreparation.

Factor A—pine ecotype:

A1—*Pinus sylvestris*;

A2—*Pinus sylvestris var. cretacea*;

Factor B—sowing term

B1—early spring;

B2—early winter;

Factor C—seed treatment, or "biopreparation"

C1—distilled water (soaking in distilled water and drying before sowing);

C2—Biogor KM (soaking in distilled water, drying and treating by finely dispersed substance before sowing);

C3—MycoCrop® (developed in Germany, CJSC Research and Production Corporation "NK Ltd.", Moscow, Russia) (soaking in distilled water, drying and applying a preparation into the soil together with the seeds).

The experimental plot size was 2 m$^2$. Number of replications—3. In each plot, 200 viable seeds (100 seeds per square meter) were sown.

No stratification was performed before early winter sowing. Before early spring sowing, the seeds were kept in the snow for a month.

The following biopreparations were used:

Distilled water (control)—to soak the seeds prior to sowing for 2 h;

BiogorKM (produced in Russia)—contained 6 strains of microorganisms (combined use of Bacillus bacteria and mycorrhizal fungi of the genus *Glomus*), their metabolites (bioactive substances), water-soluble salts of microelements catalyzing chemical transformations of exudates and products. Carrier—liquid substance. The seeds were treated by means of spraying with the finely dispersed preparation.

MycoCrop® (produced in Russia)—contained fungi *Glomus proliferum*, *G. intraradice*, *G. etunicatum*, *G. mosseae*, carrier—clay granules. The preparation was applied to the soil together with the seeds.

The seeds were sown via broadcast seeding with subsequent covering via harrowing to a depth of 1 to 2 cm. A total of 200 viable seeds per plot were sown.

The number of emerging seedlings (sprouts) and the number of surviving plants were then recorded by direct counting at each plot.

*2.3. Sampling and Analyzes*

The soil samples were collected from the 0–20 cm soil layer, ground, and sifted using a sieve with a mesh size of 1 mm. In a prepared soil sample, the following analyses were performed. The humus content was identified by using Tyurin's technique in a solution of potassium bichromate in sulfuric acid and subsequent photometric measurement at a wavelength of 590 nm https://ohranatruda.ru/upload/iblock/f09/4294828267.pdf (accessed on 18 January 2020).

The pH value was measured with a potentiometric pH-meter in an extraction of water-soluble salt. Soil to distilled water ratio was 1:5. https://ohranatruda.ru/upload/iblock/0ca/4294828015.pdf (accessed on 18 January 2020)

The total nitrogen content ($N_{total}$) was identified by using a photoelectric colorimetric technique. The wavelength was 655 nm. The thickness of cuvette (cell) was 1 cm (https://docs.cntd.ru/document/1200168815) (accessed on 18 January 2020).

The nitrate nitrogen content ($N-NO_3$) was measured by using an ionometric technique that involved nitrate extraction via alum solution (aluminium potassium sulfate) and determination by using an ion-selective technique (https://docs.cntd.ru/document/1200023499) (accessed on 18 January 2020).

The nitrification capacity of substrates was identified using Koravkov's technique. Substrate composting between the nitrate content determinations was performed at 28 °C and 60% relative humidity for 7 days (https://www.studmed.ru/aleksandrova-ln-naydenova-oa-laboratorno-prakticheskie-zanyatiya-po-pochvovedeniyu_7d8928a0579.html) (accessed on 18 January 2020).

The data obtained were processed statistically by means of two-way ANOVA or three-way analysis of variance, calculation of mean values, standard mean-square error, and the variation coefficient (relative standard deviation) in the Microsoft Excel 10 software environment. The significance level used $p < 0.05$.

The interrelation of the studied features was tested by Spearman's rank correlation ($r_s$). Standard Microsoft Excel 10 software was used for calculations. We transformed the % data prior to the ANOVA.

## 3. Results

*3.1. Scots Pine Seed Field Germination and Seedling Preservation Depending on the Sowing Terms and Biopreparations*

The early winter sowing of Scots pine seeds ensured a better germination rate and a greater seedling number compared to the early spring sowing (Table 2).

**Table 2.** Number (mean ± error) of sprouts and surviving seedlings of Scots pine under different sowing terms and different biopreparation applications for a two-year growth period on chalk outcrops.

| Sowing Term | Biopreparation | Number of Sprouts per m$^{-2}$ (Spring 2019) | | Number of First-Year Seedling per m$^{-2}$ (Autumn 2019) | | Number of Second-Year Seedling per m$^{-2}$ (Autumn 2020) | |
|---|---|---|---|---|---|---|---|
| | | M ± m | V, % | M ± m | V, % | M ± m | V, % |
| Early winter | Water | 53.8 ± 1.1 | 4.9 | 15.7 ± 0.8 | 11.9 | 12.5 ± 0.8 | 15.0 |
| | Biogor KM | 55.8 ± 0.8 | 3.5 | 17.3 ± 1.0 | 14.0 | 16.0 ± 1.3 | 20.5 |
| | MycoCrop® | 55.3 ± 0.9 | 4.1 | 16.8 ± 1.4 | 19.7 | 15.2 ± 1.2 | 18.8 |
| Early spring | Water | 41.5 ± 0.8 | 4.5 | 13.3 ± 1.1 | 21.1 | 11.3 ± 1.0 | 20.8 |
| | Biogor KM | 46.5 ± 1.8 | 9.3 | 28.5 ± 1.7 | 14.8 | 27.8 ± 1.7 | 14.8 |
| | MycoCrop® | 42.5 ± 0.8 | 4.9 | 21.7 ± 1.4 | 15.6 | 20.3 ± 1.2 | 14.2 |

Note: M—mean value; m—mean error; V—variation coefficient.

In the case of the early winter sowing, treatment by biopreparation did not appear to influence the seed field germination rate significantly. In the case of the early spring sowing, Biogor KM demonstrated a significant influence on this factor, as evidenced by the fact that the germination rate increased by 12.1%.

By autumn, the number of first-year seedlings sown in winter decreased by 70.8% in the control group. When treated with Biogor KM, by autumn, the number of seedlings decreased by 68.9%; when treated with MycoCrop®, this figure decreased by 69.6%. The difference between the experimental groups in terms of the number of surviving seedlings was insignificant at this stage.

Regarding spring sowing, the number of first-year seedlings in the control decreased by 67.9%. Upon being treated with Biogor KM in the autumn, the number of seedlings decreased by 38.7%; upon being treated with MycoCrop®, the decrease was 48.9%.

A significant difference was established in the number of surviving seedlings, depending on the pre-sowing treatment of seeds. When treated with Biogor KM, seedling survival increased by 53.3% compared to the control variant, and when treated with MycoCrop®, by 38.7%.

The results from the second year showed a significant reduction in the number of seedlings by 20.4% in the control group sown in winter. In the groups treated by biopreparations, the same trend regarding a decrease in the number of seedlings was observed.

By autumn of the second year, a significant positive effect of biopreparations on the survivability of spring-sown seedlings was observed. Given the above, the efficiency of Biogor KM in a liquid substrate used to spray the seeds was significantly higher than that of MycoCrop® on the clay substrate, which had been applied to the soil during sowing.

The difference in terms of the quantity of surviving seedlings, depending on pre-sowing treatment, was as follows: in the group treated with Biogor KM, the seedling survivability was 59.3% higher than in the control, and in the group treated once by MycoCrop®, the figure was 44.3%.

*3.2. Effect of Sowing Terms on the Seeds' Field Germination Rate and Seedling Survivability in P. sylvestris*

During the first stages of life, early winter sowing appeared to be more effective, as, at that point, the field germination rate of *P. sylvestris* seeds was 27.7% greater than in the case of early spring sowing, which, for *P. cretacea* (*P. sylvestris* var. *cretacea*), was 25.3% greater (Table 3). However, as early as by the first autumn, the number of winter-sown seedlings decreased by 72.8% in *P. sylvestris* and by 66.8% in *P. cretacea*.

**Table 3.** Number (mean ± error) of sprouts and surviving seedlings of two pine (*P. sylvestris*) ecotypes under different sowing terms and using different biopreparations for a two-year growth period on chalk outcrops.

| Ecotype | Sowing Terms | Number of Sprouts per m$^{-2}$ (Spring 2019) | | Number of First-Year Seedlings per m$^{-2}$ (Autumn 2019) | | Number of Second-Year Seedlings per m$^{-2}$ (Autumn 2019) | |
|---|---|---|---|---|---|---|---|
| | | **M ± m** | **V, %** | **M ± m** | **V, %** | **M ± m** | **V, %** |
| *P. sylvestris* | Winter | 54.0 ± 0.9 | 4.8 | 14.7 ± 0.3 | 6.8 | 12.4 ± 0.5 | 12.8 |
| | spring | 42.3 ± 0.6 | 4.6 | 18.8 ± 2.2 | 35.6 | 17.6 ± 2.3 | 39.6 |
| *P. cretacea* | Winter | 56.0 ± 0.5 | 2.8 | 18.6 ± 0.7 | 11.1 | 16.7 ± 0.8 | 15.3 |
| | spring | 44.7 ± 1.5 | 10.1 | 23.6 ± 2.4 | 30.6 | 22.1 ± 2.6 | 35.5 |

Note: M—mean value; m—mean error; V—variation coefficient.

*P. cretacea* seedlings showed greater survivability in comparison to that of *P. sylvestris* in the second year of life, both for winter and summer sowing.

When early spring sowing was practiced, the number of seedlings in autumn decreased by 55.6% in the *P. sylvestris* group and by 47.2% in the *P. cretacea* group. By the end of the first year of life, the spring-sown seedlings had greater survivability than those sown in early winter, by 21.8% and 21.2%, respectively, with variation coefficients of the trait exceeding 30%. In the second year of life, in the *P. sylvestris* group, winter-sown seedlings demonstrated an autumn death rate of 15.6% compared to the previous period; in the *P. cretacea* group, the respective factor equaled 10.2%. The number of spring-sown second-year seedlings did not change significantly, as evidenced by a relatively high variance coefficient of the trait—over 35%.

### 3.3. Effect of Biopreparations in the Seeds Field Germination Rate and Seedlings Survivability in the Two Studied Ecotypes of P. sylvestris

Treating seeds with Biogor KM and MycoCrop® did not significantly influence the field germination rate of the studied seeds. However, a certain positive trend for this trait was seen in both ecotypes (Table 4).

**Table 4.** Number (mean ± error) of sprouts and number of surviving seedlings in two studied ecotypes of *P. sylvestris* treated by different biopreparations for a two-year growth period on chalk outcrops.

| Ecotype | Biopreparation | Number of Sprouts per m$^{-2}$ (Spring 2019) | | Number of First-Year Seedlings per m$^{-2}$ (Autumn 2019) | | Number of Second-Year Seedlings per m$^{-2}$ (Autumn 2020) | |
|---|---|---|---|---|---|---|---|
| | | **M ± m** | **V, %** | **M ± m** | **V, %** | **M ± m** | **V, %** |
| *P. sylvestris* | Water | 46.7 ± 2.7 | 14.0 | 12.7 ± 0.8 | 15.5 | 10.2 ± 0.5 | 11.6 |
| | BiogorKM | 49.7 ± 2.7 | 13.5 | 20.5 ± 2.4 | 28.4 | 19.2 ± 2.7 | 34.6 |
| | MycoCrop® | 48.2 ± 2.8 | 14.1 | 17.0 ± 1.5 | 21.7 | 15.7 ± 1.3 | 20.4 |
| *P. cretacea* | Water | 48.7 ± 3.1 | 15.5 | 16.3 ± 0.7 | 10.1 | 13.7 ± 0.4 | 7.7 |
| | BiogorKM | 52.7 ± 2.0 | 9.4 | 25.3 ± 2.9 | 28.1 | 24.7 ± 2.9 | 28.5 |
| | MycoCrop® | 49.7 ± 3.2 | 15.7 | 21.5 ± 1.3 | 15.2 | 19.8 ± 1.4 | 17.3 |

Note: M—mean value; m—mean error; V—variation coefficient.

By autumn, a trend indicating better survivability in the first-year seedlings was more pronounced in the *P. cretacea* group than in the *P. sylvestris* group, given both biopreparations were equally applied. By autumn, the number of first-year *P. sylvestris* seedlings treated with Biogor KM decreased by 58.7%, and those treated by MycoCrop® decreased by 64.7%. The survivability of *P. sylvestris* seedlings in this experimental group appeared to be 61.4% and 33.8% higher, respectively, compared to the control one. In the *P. cretacea* group, the

number of first-year seedlings treated with Biogor KM decreased by 51.9% in autumn and, when treated by MycoCrop®, by 64.7%. The survivability of *P. cretacea* seedlings in this experimental group was higher than in the control group by 55.2% and 31.9%, respectively.

Certain differences between studied pine ecotypes in terms of stand survivability were indicated by the first autumn, and they apparently depended on the treatment type. In the control group, the number of *P. cretacea* seedlings was 28.3% greater than in the *P. sylvestris* group. Upon seed treatment, the above tendency continued as follows: in the case of treatment by Biogor KM, the number of *P. cretacea* seedlings was 23.4% more than that of *P. sylvestris* seedlings; in the case of treatment by MycoCrop®, the figure was 26.5% higher.

By the second autumn, stand survivability in the control groups deteriorated, and the number of seedlings decreased by 19.7% in *P. sylvestris* and by 15.9% in *P. cretacea*. The differences between ecotypes in terms of stand survivability were determined. In the control, the number of *P. cretacea* seedlings was 34.3% higher than that of *P. sylvestris*. When the seeds had been treated by Biogor KM, the number of *P. cretacea* seedlings was 28.6% greater than that of *P. sylvestris;* when treated by MycoCrop®, the figure was 26.1%.

### 3.4. Three-Way ANOVA of the Seed Germination Rate and Seedling Survivability Variance

The significant effect of the main factors on the absolute survivability of pine seedlings grown on chalk outcrops at a probability level $p = 0.05$ was revealed (Table 5).

**Table 5.** The estimation of the organized factors' effect size on the total number of seedlings in two *P. sylvestris* ecotypes on chalk outcrops obtained by the three-way analysis of variance (2019–2020).

| Source of Variation | Number of Sprouts (Spring 2019) | | Number of First-Year Seedlings (Autumn 2019) | | Number of Second-Year Seedlings (Autumn 2020) | |
|---|---|---|---|---|---|---|
| | $F_f$ | $h^2_x$ | $F_f$ | $h^2_x$ | $F_f$ | $h^2_x$ |
| Organized factors, total | 21.6 | 90.8 * | 33.0 | 93.8 * | 33.0 | 93.8 * |
| A | 7.4 | 2.8 * | 36.7 | 14.4 * | 48.8 | 12.6 * |
| B | 208.0 | 79.4 * | 40.5 | 15.9 * | 70.5 | 18.2 * |
| C | 6.6 | 5.0 * | 46.3 | 36.4 * | 85.2 | 44.0 * |
| A × B | 0.0 | 0.0 | 0.4 | 0.2 | 0.1 | 0.01 |
| A × C | 0.3 | 0.2 | 0.2 | 0.2 | 0.9 | 0.5 |
| B × C | 1.9 | 1.4 | 29.7 | 23.3 * | 35.7 | 18.4 * |
| A × B × C | 2.5 | 1.9 | 0.4 | 0.3 | 0.1 | 0.01 |

Note: Factor A—"pine ecotype"; Factor B—"sowing term"; Factor C—"biopreparation", $h^2_x$—size of effect on the effective feature; $F_r$—F-ratio (Fisher's criterion); *—the influence of the factor is significant at $p = 0.05$.

The main factor with the greatest effect on seed germination and survivability at the first stage of growth is the "sowing term," accounting for 79.4% of the effective feature's (number of seedlings) total variance. A null hypothesis of the pine ecotype effect at all stages of the seedling's growth should be rejected. As the seedlings' age progresses, a share of the "ecotype" factor effect on the survivability increases from 7.4% at the sprouting stage to 12.6%–14.4% at the end of the first and the second vegetation season, respectively. The share of the "biopreparation" factor effect appears to grow up to 44.0% by the second year of life and accounts for the greatest share in the total variance of the resulting feature "number of seedlings per m$^{-2}$". As the age increases, the share of interacting factors (e.g., "sowing term"—"biopreparation") in the total variance seems to grow from insignificant values to 18.4%.

### 3.5. The Effect of Pine Ecotypes, Sowing Terms, and Biopreparations on Seedling Survivability in P. sylvestris at Chalk Outcrops

The survivability of seedlings during the four most critical growth periods was assessed. The number of surviving plants from each previous period was taken for 100%. The results of the assessment of each studied factor's effect (pine ecotype, sowing term, and biopreparation) on the relative value of the seedling's survivability for various growth periods are presented in Table 6.

**Table 6.** Survivability (mean ± error) of *P. sylvestris* seedlings (%) on chalk outcrops depending on the pine ecotype, sowing term, and biological preparations in different growth periods (each preceding period is taken for 100%).

| Factors | Developmental Periods | | | | | | | |
|---|---|---|---|---|---|---|---|---|
| | 1st Period | | 2nd Period | | 3rd Period | | 4th Period | |
| | M ± m | V, % | M ± m | V, % | M ± m | V, % | M ± m | V, % |
| *Pine ecotype (Factor A)* | | | | | | | | |
| *P. sylvestris* | 48.2 ± 1.5 | 13.3 | 35.6 ± 3.2 | 37.6 | 91.7 ± 1.8 | 8.2 | 96.5 ± 1.3 | 5.7 |
| *P. cretacea* | 50.3 ± 1.6 | 13.3 | 42.6 ± 3.1 | 30.6 | 91.7 ± 1.4 | 6.5 | 99.2 ± 0.5 | 2.2 |
| *Sowing term (Factor B)* | | | | | | | | |
| Early winter | 55.0 ± 0.5 | 4.2 | 30.1 ± 0.9 | 13.0 | 90.3 ± 1.8 | 8.5 | 96.7 ± 1.3 | 5.5 |
| Early spring | 43.5 ± 0.8 | 8.2 | 48.1 ± 3.2 | 28.6 | 93.1 ± 1.2 | 5.7 | 99.1 ± 0.7 | 2.8 |
| *Biological preparation (Factor C)* | | | | | | | | |
| Water | 47.7 ± 2.0 | 14.2 | 30.6 ± 1.5 | 17.0 | 86.5 ± 1.4 | 5.7 | 95.3 ± 1.6 | 5.8 |
| Biogor KM | 51.2 ± 1.7 | 11.4 | 46.1 ± 4.7 | 35.6 | 95.3 ± 2.1 | 7.6 | 99.4 ± 0.6 | 1.9 |
| MycoCrop® | 48.9 ± 2.0 | 14.3 | 40.6 ± 3.5 | 29.9 | 93.3 ± 1.2 | 4.6 | 98.9 ± 1.1 | 3.9 |

Note: M—mean value; m—mean error; V—variation coefficient. The 1st period—sprouts (2019); 2nd period: sprouts—wintering in the first year of life (2019); 3rd period: wintering in the first year of life—springtime vegetation renewal (2020); 4th period—springtime vegetation renewal—wintering in the second year (2020).

If sown directly, the *P. cretacea* ecotype was found to have great potential in terms of survivability compared to the *P. sylvestris* ecotype. *P. cretacea* seedlings demonstrated significantly greater survivability by the winter of the second year, both during the first year of life and in the long run. Sowing *P. sylvestris* seeds in winter ensures a better germination rate and survivability of the stand in the year of sowing. However, our data indicate that the surviving plants in the second year (3rd and 4th periods) showed greater survivability and viability as opposed to the plants grown from winter-sown seeds.

Applying biopreparations during sowing markedly increased the seedling survivability during all growth periods. Using Biogor KM preparation to treat the seeds has been proven to be more effective than applying MycoCrop® in the soil together with the seeds.

The three-way variance analysis showed that all investigated organized factors had a significant influence on the "seedling survivability" effective feature during all growth periods of the plants (Table 7).

It was revealed that, during the initial stages of plant development in a year of sowing, the greatest significant factor was the "sowing terms" ($h^2_x = 79.4\%$), although the other studied main factors are also significant. At later stages, while the involvement of random factors increases (by around 50%), the effect of "pine ecotype" and "biopreparation" factors significantly increases up to $h^2_x = 10.1$ and $h^2_x = 18.4$, respectively. Table 8 presents a calculation of the Spearman's rank correlation between soil agrochemical properties and stand survivability in different growth periods. The correlation between the soil substrate's nitrifying capacity and seedling survivability in the first three periods ($r_s = 0.617$–$0.673$, $p < 0.05$) was moderate (Table 8).

**Table 7.** The assessment of the effect of organized factors on *P. sylvestris* seedling survivability during different periods of growth on chalk outcrops, obtained by the three-way analysis of variance (each preceding period is taken for 100%).

| Source of Variation | Percentage of Surviving Seedlings during, % | | | | | | | |
| --- | --- | --- | --- | --- | --- | --- | --- | --- |
| | 1st Period | | 2nd Period | | 3rd Period | | 4th Period | |
| | $F_f$ | $h^2_x$ | $F_f$ | $h^2_x$ | $F_f$ | $h^2_x$ | $F_f$ | $h^2_x$ |
| Organized factors, total | 21.63 | 90.84 * | 35.64 | 94.23 * | 2.2 | 50.1 * | 2.2 | 50.3 * |
| Factor A | 7.4 | 2.8 * | 28.7 | 6.9 * | 0.0 | 0.0 | 4.9 | 10.1 |
| Factor B | 208.0 | 79.4 * | 188.8 | 45.4 * | 2.2 | 4.7 | 3.8 | 7.8 |
| Factor C | 6.6 | 5.0 * | 48.3 | 23.2 * | 7.5 | 32.5 * | 4.4 | 18.4 * |
| A × B | 0.0 | 0.0 | 0.7 | 0.2 | 0.6 | 2.3 | 3.7 | 7.5 |
| A × C | 0.3 | 0.2 | 0.3 | 0.2 | 1.1 | 4.9 | 0.8 | 3.3 |
| B × C | 1.9 | 1.4 | 37.1 | 17.8 * | 0.3 | 1.2 | 0.4 | 1.7 |
| A × B × C | 2.5 | 1.9 | 1.2 | 0.6 | 1.0 | 4.6 | 0.4 | 1.6 |

Note: The 1st period—sprouts (2019); 2nd period: sprouts—wintering in the first year of life (2019); 3rd period: wintering in the first year of life—springtime vegetation renewal (2020); 4th period—springtime vegetation renewal—wintering in the second year (2020); *—the influence of the factor is significant at *p* = 0.05. Factor A—"pine ecotype"; Factor B—"sowing term"; Factor C—"biopreparation", $h^2_x$—size of effect on the effective feature; $F_r$—F-ratio (Fisher's criterion).

**Table 8.** The correlation between chemical and biological soil properties and *P. sylvestris* stand survivability at different growth periods.

| Chemical and Biological Soil Properties | *P. sylvestris* Seedlings Survivability at Different Growth Periods, % | | | |
| --- | --- | --- | --- | --- |
| | 1st Period | 2nd Period | 3rd Period | 4th Period |
| Nitrifying capacity, mg kg$^{-1}$ | 0.627 | 0.673 | 0.617 | 0.298 |
| N-NO$_3$ content, mg kg$^{-1}$ | 0.473 | 0.399 | 0.396 | −0.091 |
| N$_{total}$ content, % | 0.212 | 0.203 | 0.002 | 0.152 |
| Humus content, % | 0.183 | 0.051 | 0.047 | 0.171 |
| pH | 0.270 | −0.043 | 0.201 | −0.068 |

No correlation was found between the stand survivability at different growth periods and N$_{total}$, humus content in the soil, and the soil pH.

## 4. Discussion

The present study highlights ways to enhance the efficiency of re-afforestation measures by sowing seeds on chalk outcrops using two Scots pine ecotypes. The influence of biopreparations based on a fungal consortium from the order *Gnomales*, *Bacillus* bacteria, and bioactive substances on the growth of Scots pine seedlings depending on the ecotype is discussed.

As far as the growth and development of pine seedlings is concerned, four crucial periods were revealed in the first two years of life.

The first one is "sowing—sprouts". The seed germination period is considered the most vulnerable in terms of biotic and abiotic stressors. It is susceptible to unpredictable environmental impacts (including factors such as temperature, humidity, and light) [55–57].

Among all of the above listed abiotic factors, temperature appears to be the critical one. Temperature directly affects biochemical reactions during seed germination in case no water deficit is observed [56,58]. The optimal temperature for seed germination is identified, on the one hand, by the environmental conditions in which the parent plants are growing, and on the other hand, by the conditions in which the seeds reach maturity, and it appears

to influence the time and pace of this process [59–61]. The germination percentage and the time of germination are the two most important traits of seed germination capacity. Firstly, they determine the number of germinated seeds and, secondly, the number of surviving and persisting seedlings [62–64].

In the conditions of the chalk outcrops, seed germination is a fundamental period for re-afforestation and woodland formation by direct sowing and provides grounds for spontaneous self-seeding in forest plantations. Together with the substrate properties mentioned beforehand, such as high mobility, cobble content, pH, high surface albedo, temperature and water regime, and the absence of snow cover in winter, which may negatively affect seed germination and stand survivability, one more negative factor can be observed—soil heaving.

A "soil heaving" phenomenon results from intense cryogenic reactions in the chalky substrate; they are caused by changes in temperature gradients. Ice crystals, while forming, cause periodic rises and falls in the upper 2–5 cm layer (Figure 4).

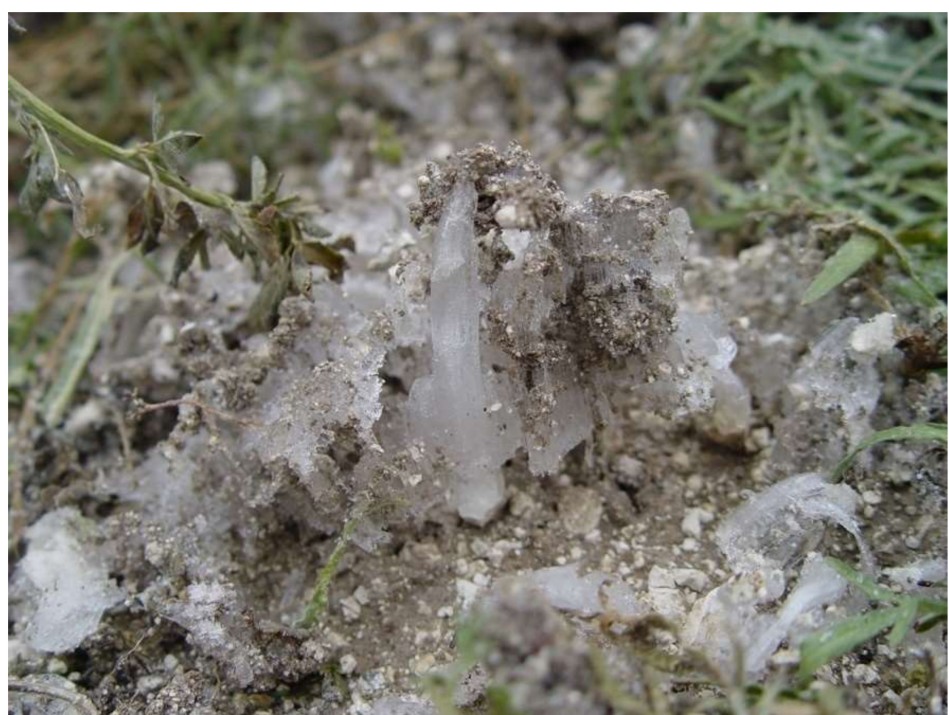

**Figure 4.** The result of the cryogenic process "soil heaving" in the surface layer of the substrate of chalk outcrops (photo by V.I. Chernyavskikh).

These reactions were frequently noticed by researchers both in autumn and in spring during spring frosts [65]. This process appears to influence the reforestation efforts on chalk outcrops by the direct sowing and self-seeding of pine trees, but its effect is ambivalent. Positive effects of this phenomenon include a possibility of spontaneous unassisted seed covering at a certain depth in case of surface planting without any additional cultural practices. A negative aspect is the chance of seeds ending up at a greater depth (over 4–5 cm), which makes it impossible for them to sprout and emerge or results in weakened stands. Additionally, the death of already existing seedlings may occur due to mechanical damage.

During the growth stage ("sprouts—wintering in the first year of life" on a chalk outcrop), insolation is the greatest danger, which may lead to plant death because of anomalously high temperatures and albedos. Sun rays, particularly those of the infrared spectrum variety, when reflecting from the light surface of a chalky outcrop, may cause needle burning in the lower part or drying of the entire plant. The calculation of the plants' survivability rate in this period allowed us to evaluate the resistance of the first-year seedlings to a set of adverse environmental conditions in the summertime.



The "wintering in the first year of life—springtime vegetation renewal" period helped us to evaluate the first-year seedlings' resistance to the following winter hardiness factors: winter drying, frosts, substrate blow-ups, etc.

The "springtime vegetation renewal—wintering in the second year" period allowed the evaluation of the second-year plants' resistance to adverse summertime weather conditions.

It was revealed that the biopreparations positively affect the field germination rate and seedling survivability at the first stages of life in both *P. sylvestris* ecotypes. The bacteria and mycorrhizal fungi combinations are of special importance in relation to the soil and the seed treatment. The bacteria and mycorrhizal fungi combinations were proven to have a powerful synergic effect. In the soil, their symbiotic relationships promote each other's activity [47,48].

During the experiment, the substrate nitrification capacity increased. This is indirect proof of the greater biological activity in the rhizosphere. The above revealed a positive correlation between the nitrification capacity level and seedling survivability during the first years of tree growth, providing evidence of the positive effect on the soil biological process induced by the investigated biopreparations based on the microorganismal consortium.

The selection of efficient microorganisms and the seeds' thin film coating provided protection against soil pathogens after sowing. The roots and mycorrhiza were enveloped and protected by a microbial consortium, subsequently forming the rhizosphere [66]. The consortium, obtaining sugars and performing protective functions more intensively, e.g., producing antibiotics as well as decomposing organic matter, being involved in the biogeochemical cycle, provides plant nutrition [67,68]. The only drawback of biopreparations is that their effect cannot be proven to be stable from year to year and in the long term [69].

The bacteria are assigned a key role in atmospheric nitrogen uptake and phosphorus solubilization [70]. Nitrogen-fixing microorganisms make it possible to supply plants with available nitrogen and increase soil fertility. The metabolites promote a root system development. Phosphorolytic bacteria transform complex organophosphates and mineral phosphates into plant-available forms [71–73]. To facilitate the growth and productivity of young and mature Scots pine stands, enhance the formation and stability of soil organic matter [74], improve mycorrhizal and rhizobium inoculation with organic and inorganic fertilizers [75], and passivate heavy metals [76], the Biogeosystem Technique (BGT*) methodology was developed [76]. The BGT* methodology includes the following fundamentally new transcendental scientific and technological capabilities: intrasoil milling [77], intrasoil pulse continuous-discrete watering [78], and the intrasoil application of dispersed matter, including stimulants and nanoparticles [79–81] during intrasoil milling and/or intrasoil pulse continuous-discrete watering to improve the soil and stimulate the tree growth.

Numerous studies are being conducted on the symbiosis of woody plants and various groups of microorganisms on different soils. Of particular interest are data on the increase in the proportion of nitrogen-fixing trees on alkaline soils, as well as in temperate areas with a decrease in precipitation [82,83]. Data on the symbiosis of woody plants associated with groups of bacteria (rhizobia, actinobacteria, cyanobacteria) have been published, and corresponding databases have been prepared [84]. These data are essential for understanding the results of our research and for the further use of biological methods to increase the productivity and sustainability of tree crops.

Studies on the influence of bacteria and their consortiums on the most important soil processes in forest ecosystems have advanced greatly in recent years and are well documented. Bacteria take part in the decomposition of the dead organic matter of plants and mycelium of dead fungi. They perform a protective function under the action of trace elements and other inorganic pollutants on the root systems of plants due to the processes of biosorption, bioaccumulation, and biotransformation in polluted habitats [85,86]. An understanding of a forest communities soil bacterial ecology is important. A new level understanding with regard to bacteria and their consortia contribution to soil processes in forest crops and ecosystems is needed [87]. A woody vegetation accounts for the majority

of the planet's total vegetation biomass [88]. In this regard, the biologically active consortia of fungi and bacteria and BGT* methodology usage in reforestation activities acquire ecological and environmental significance.

## 5. Conclusions

Chalk outcrops as a substrate appear to possess a number of negative properties that require a complex approach to re-afforestation measures based on sowing *P. sylvestris*, including the following: different sowing terms, planting local ecotypes, treating seeds with biopreparations based on microorganismal consortia and bioactive substances.

Biopreparations based on the consortium of *Gnomales* fungi, *Bacillus* bacterium, and bioactive substances are highly effective, biologically speaking, during the seed reproduction of Scots pine and forest renewal efforts involving this pine on carbonate outcrops.

The application of Biogor KM, a commercially manufactured biopreparation, on a liquid carrier for pre-sowing seed treatment and clay-based MycoCrop® preparation for pre-sowing application in the soil together with the seeds enhances their germination rate and increases the survivability of Scots pine seedlings (in the case of the early spring sowing). Using these preparations during winter sowing was not found to demonstrate any proven positive effect.

It is advisable to develop forest plantations on chalk outcrops using the Scots pine ecotype *P. sylvestris* var. *cretacea*, treating its seeds via soaking in a consortium-based biopreparation and with Biogor KM before sowing. To improve the growth and productivity of young and mature Scots pine stands, the Biogeosystem Technique (BGT*) methodology was developed.

**Author Contributions:** Conceptualization, V.M.K.; data curation, V.I.C.; formal analysis, E.V.D.; funding acquisition, A.P.G.; investigation, V.I.C., L.D.S. and H.V.G.; methodology, A.A.Z.J. and V.I.C.; project administration, V.I.C.; resources, V.I.C.; software, L.D.S.; supervision, V.M.K.; validation, N.A.S. and V.P.K.; visualization, V.I.C. and E.V.D.; writing—original draft, E.V.D.; writing—review and editing, S.V.A., L.V.P., S.V.K. and V.P.K. All authors have read and agreed to the published version of the manuscript.

**Funding:** The research was carried out within the framework of the State task of the Ministry of Education on the topic: "Immobilization of trace elements by the products of interactions of layered silicates with soil organic matter and microorganisms" (Additional Agreement No. 073-03-2023-030/2 from 14 February 2023 to Agreement № 073-00030-23-02 from 13 February 2023).

**Data Availability Statement:** Not applicable.

**Conflicts of Interest:** The authors declare no conflict of interest.

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
