# Peer review of "Using Microorganismal Consortium and Bioactive Substances to Treat Seeds of Two Scots Pine Ecotypes as a Technique to Increase Re-Afforestation Efficiency on Chalk Outcrops"

_forests, doi:10.3390/f14061093_

Round 1
Reviewer 1 Report
Kindly check the Attatched file.

Author Response
Dear Reviewer 1,
Thank you for your encouraging review.
Our answers are in Italics.
Dear Authors, Congratulations. It is well studied work. The title “Using microorganismal consortium and bioactive substances to treat seeds of two scots pine ecotypes as a technique to increase reafforestation efficiency on chalk outcrops.” fits the profile of “Forests” journal. The study delivers some interesting results and can be a source of valuable information.
However, the authors made shortcomings that should be corrected and/or revised before the publication of this work.
Abstract
Kindly check minor spelling mistake. The abstract needs a better composition of words.
Abstract corrected
Introduction
Authors should explain why they chose two scots pine in this work.
The Pinus sylvestris L. is used for a creation of forest plantations in the complex landscape conditions of the carbonate soils on the chalk outcrops. Two main Pinus sylvestris ecotypes grow in the south of the Central Russian Upland. An ecotype P. sylvestris L. is the most widely used in cultivation. A second ecotype Pinus sylvestris var. cretacea Kalenicz. is an indigenous relict species growing on chalk outcrops. The studies examined both ecotypes in order to identify a species most promising for cultivation on Cretaceous outcrops. A detailed description of ecotypes is given below.
Material and methods
Insert the area studied in this experiment if possible.
The total accounting area of the experimental plots was 42 m2. The total area of the experiment was 168 m2.
Results
Results are well indicated.
Than you one more time.
Discussion
Could be more detailed. The lines should be more focused for highlighting the important finding of this work.
Numerous studies are being conducted on the symbiosis of woody plants and various groups of microorganisms on different soils. Of particular interest are data on an increase in the proportion of nitrogen-fixing trees on alkaline soils, as well as in temperate areas with a decrease in precipitation [82,83]. Data on the symbiosis of woody plants associated with groups of bacteria (rhizobia, actinobacteria, cyanobacteria) have been published and corresponding databases prepared [84]. These data are essential for understanding the results of our research and further use of biological methods to increase the productivity and sustainability of tree crops.
The studies on an influence of bacteria and their consortiums on the most important soil processes in forest ecosystems has advanced greatly in recent years and are well documented. Bacteria take part in a decomposition of a dead organic matter of plants and a mycelium of dead fungi. An understanding of a forest communities soil bacterial ecology is important. A new level understanding is needed of a bacteria and their consortia contribution to soil processes in the forest crops and ecosystems [85].
A woody vegetation accounts for a majority of the planet's total vegetation biomass [86]. In this regard, the biologically active consortia of fungi and bacteria and BGT* methodology usage in reforestation activities acquires ecological and environmental significance.
References
Strengthen the part of the discussion with 2 or 3 new references.
- Steidinger, B. S., Crowther, T. W., Liang, J., Van Nuland, M. E., Werner, G. D. A., Reich, P. B., Nabuurs, G. J., de-Miguel, S., Zhou, M., Picard, N., Herault, B., Zhao, X., Zhang, C., Routh, D., & Peay, K. G. Climatic controls of decomposition drive the global biogeography of forest-tree symbioses. Nature 2019, 569, 404–408. DOI: https://doi.org/10.1038/s41586-019-1128-0
- Gei, M., Rozendaal, D. M. A., Poorter, L., Bongers, F., Sprent, J. I., Garner, M. D., Aide, T. M., Andrade, J. L., Balvanera, P., Becknell, J. M., Brancalion, P. H. S., Cabral, G. A. L., César, R. G., Chazdon, R. L., Cole, R. J., Colletta, G. D., de Jong, B., Denslow, J. S., Dent, D. H., ... Powers, J. S. Legume abundance along succes-sional and rainfall gradients in Neotropical forests. Nature Ecology and Evolution. 2018, 2, 1104–1111. DOI: https://doi.org/10.1038/s41559-018-0559-6
- Tedersoo, L., Laanisto, L., Rahimlou, S., Toussaint, A., Hallikma, T., & Pärtel, M. Global database of plants with root-symbiotic ni-trogen fixation: NodDB. Journal of Vegetation Science 2018, 29, 560–568. DOI: https://doi.org/10.1111/jvs.12627
- Lladó S., López-Mondéjar R., Baldrian P. Soil Bacteria: Diversity, Involvement in Ecosystem Processes, and Response to Global Change. Microbiology and Molecular Biology Reviews 2017, 81(2), 12. DOI: https://doi.org/10.1128/MMBR.00063-16
- Erb, K.-H., Kastner, T., Plutzar, C., Bais, A. L. S., Carvalhais, N., Fetzel, T., Gingrich, S., Haberl, H., Lauk, C., Niedertscheider, M., Pongratz, J., Thurner, M., & Luyssaert, S. Unexpectedly large impact of forest management and grazing on global vegetation biomass. Nature, 2018, 553, 73–76. DOI: https://doi.org/10.1038/nature25138
Conclusion
Minor language issues must be addressed to improve quality of the MS
Done
Your comments and suggestions were helpful.
Thank you.
From the name of the authors,
Regards,
Valery P. Kalinitchenko
Reviewer 2 Report
My commemts and questions are in attached article file as a note box and highlighted with the yellow and pink color.
Due to not having a line number it is dificult to list the points.
Title, subtitles and subheadings, Please use uppercase for each word
of the genus
The study revealed that biopreparations and bioactive substances promote higher pine seed germination rates, and ensure seedlings stability and survivability in unfavorable plant and tree organogenesis conditions.
The afforestation of the slopes, and abandoned and degraded farmlands provide ecosystem sustainable development. This is an important tool in land resource management,
[4-8]
[9-12]
underway
was the
in the Belgorod region.
landscapes
areas
the widest
The ability
interaction
seed
Methods
Please edit the font size
Please clarify the software name used for data analysis and ANOVA. Please mention if you transformed any of the % data such as germination% or % seedlings survival prior to teh ANOVA
a key role in atmospheric
forms
a commercially manufactured
Please follow the journal guidlines for preparation of the references list

My commemts and questions are in attached article file as a note box and highlighted with the yellow and pink color.
Due to not having a line number it is dificult to list the points.
Title, subtitles and subheadings, Please use uppercase for each word
of the genus
The study revealed that biopreparations and bioactive substances promote higher pine seed germination rates, and ensure seedlings stability and survivability in unfavorable plant and tree organogenesis conditions.
The afforestation of the slopes, and abandoned and degraded farmlands provide ecosystem sustainable development. This is an important tool in land resource management,
[4-8]
[9-12]
underway
was the
in the Belgorod region.
landscapes
areas
the widest
The ability
interaction
seed
Methods
Please edit the font size
Please clarify the software name used for data analysis and ANOVA. Please mention if you transformed any of the % data such as germination% or % seedlings survival prior to teh ANOVA
a key role in atmospheric
forms
a commercially manufactured
Please follow the journal guidlines for preparation of the references list
Author Response
Dear Reviewer 2,
Thank you for your comments. We accepted your propositions with gratitude.
The text of the manuscript was corrected according to the pdf-file you attached to your review.
For the data analysis and ANOVA, we used the Excel 10 program product. We transformed the % data prior to the ANOVA.
Special thanks for your English language corrections.
From the name of the authors,
Regards,
Valery P. Kalinitchenko